# Lafora Disease: A Case Report and Evolving Treatment Advancements

**DOI:** 10.3390/brainsci13121679

**Published:** 2023-12-06

**Authors:** Carola Rita Ferrari Aggradi, Martina Rimoldi, Gloria Romagnoli, Daniele Velardo, Megi Meneri, Davide Iacobucci, Michela Ripolone, Laura Napoli, Patrizia Ciscato, Maurizio Moggio, Giacomo Pietro Comi, Dario Ronchi, Stefania Corti, Elena Abati

**Affiliations:** 1Dino Ferrari Centre, Department of Pathophysiology and Transplantation (DEPT), University of Milan, 20122 Milan, Italy; carola.ferrari@unimi.it (C.R.F.A.); gloria.romagnoli@unimi.it (G.R.); megi.meneri@unimi.it (M.M.); giacomo.comi@unimi.it (G.P.C.); dario.ronchi@unimi.it (D.R.); 2Neuromuscular and Rare Diseases Unit, Department of Neuroscience, Fondazione IRCCS Ca’ Granda Ospedale Maggiore Policlinico, 20122 Milan, Italy; martina.rimoldi@policlinico.mi.it (M.R.); daniele.velardo@policlinico.mi.it (D.V.); michela.ripolone@policlinico.mi.it (M.R.); patrizia.ciscato@policlinico.mi.it (P.C.); maurizio.moggio@policlinico.mi.it (M.M.); 3Medical Genetics Unit, Fondazione IRCCS Ca’ Granda Ospedale Maggiore Policlinico, 20122 Milan, Italy; 4Stroke Unit, Fondazione IRCCS Ca’ Granda Ospedale Maggiore Policlinico, 20122 Milan, Italy; 5Neurology Unit, Fondazione IRCCS Ca’ Granda Ospedale Maggiore Policlinico, 20122 Milan, Italy

**Keywords:** Lafora disease, therapeutic strategies, EPM2B, EPM2A, tonic–clonic seizures, Lafora bodies, laforin, malin

## Abstract

Lafora disease is a rare genetic disorder characterized by a disruption in glycogen metabolism. It manifests as progressive myoclonus epilepsy and cognitive decline during adolescence. Pathognomonic is the presence of abnormal glycogen aggregates that, over time, produce large inclusions (Lafora bodies) in various tissues. This study aims to describe the clinical and histopathological aspects of a novel Lafora disease patient, and to provide an update on the therapeutical advancements for this disorder. A 20-year-old Libyan boy presented with generalized tonic–clonic seizures, sporadic muscular jerks, eyelid spasms, and mental impairment. Electroencephalography showed multiple discharges across both brain hemispheres. Brain magnetic resonance imaging was unremarkable. Muscle biopsy showed increased lipid content and a very mild increase of intermyofibrillar glycogen, without the polyglucosan accumulation typically observed in Lafora bodies. Despite undergoing three lines of antiepileptic treatment, the patient’s condition showed minimal to no improvement. We identified the homozygous variant c.137G>A, p.(Cys46Tyr), in the *EPM2B/NHLRC1* gene, confirming the diagnosis of Lafora disease. To our knowledge, the presence of lipid aggregates without Lafora bodies is atypical. Lafora disease should be considered during the differential diagnosis of progressive, myoclonic, and refractory epilepsies in both children and young adults, especially when accompanied by cognitive decline. Although there are no effective therapies yet, the development of promising new strategies prompts the need for an early and accurate diagnosis.

## 1. Introduction

Lafora disease (LD, OMIM #254780) is a rare, autosomal, recessive, neurodegenerative disorder, belonging to a group of epilepsies defined as progressive myoclonus epilepsies (PMEs) [1]. It has an estimated prevalence of approximately four cases per one million individuals and it occurs most frequently in Mediterranean countries, South India, North Africa, and the Middle East [2]. Early LD symptoms may appear during late childhood or adolescence and typically include myoclonus, visual seizures, hallucinations, generalized tonic–clonic seizures, muscle wasting, behavioral changes, dysarthria, depression, and cognitive decline [3,4,5]. The clinical phenotype invariably worsens over time, resulting in a fatal outcome within 10 years of symptom onset [6].

LD is characterized by a pathognomonic feature: the presence of Lafora bodies (LBs) in the brain and other tissues. LBs are inclusions resulting from the build-up of polyglucosans, which are insoluble glycogen molecules characterized by abnormally elongated and flawed structures, lacking the typical spherical shape.

Despite LBs being considered a hallmark of the disease, genetic testing is required to confirm the diagnosis in a patient presenting with LD symptomatology. Most LD cases (approximately 90%) can be ascribed to pathogenic variants occurring along *EPM2A* [7] or *EPM2B/NHLRC1* [8] genes. Together, the proteins encoded by such genes—the glucan phosphatase laforin (encoded by *EPM2A*) and the E3-ubiquitin ligase malin (encoded by *EPM2B*)—form the so-called laforin–malin complex, which has the function of negatively regulating glycogen synthesis as well as improving glycogenolysis [2]. Molecular defects in either *EPM2A* or *EPM2B*, by altering either laforin or malin individually, are also able to affect the general function of this complex, therefore resulting in glycogen accumulation. For this reason, patients carrying mutations in these genes can display similar clinical phenotypes, although laforin and malin play different roles as single proteins [2,8,9,10,11,12]. In *EPM2A*- and *EPM2B*-mutated patients, glycogen forms inadequately branched and elongated chains, leading to the formation of LBs, with subsequent neuroinflammation, neurodegeneration, and epilepsy [13]. To date, over 150 defects have been identified in the *EPM2A* and the *EPM2B* genes, in more than 250 LD cases [13,14,15,16,17].

In this report, we present the clinical and myopathological report of a 20-year-old boy with progressive myoclonus epilepsy, diagnosed with LD, and displaying unusual lipid inclusions at muscle biopsy. The patient went undiagnosed for 10 years before receiving the LD diagnosis in our center. The homozygous likely pathogenic variant c.137G>A, p.(Cys46Tyr), was identified in the *EPM2B* gene, by performing targeted next-generation (clinical exome) sequencing.

## 2. Patient and Methods

This study was approved by the institutional review board of the Fondazione IRCCS Ca’ Granda Ospedale Maggiore Policlinico. Written informed consent was obtained from the patient’s caregivers. Experimental methods are described in Appendix A.

A 20-year-old Libyan boy was referred to our Neurology Unit, due to worsening symptoms such as seizures, eyelid twitches, and cognitive decline. He was the seventh born to healthy consanguineous parents, after an uneventful pregnancy. No significant family history was reported. He had regular psychomotor development and no other relevant medical conditions. When he was 10 years old, he developed subtle myoclonic movements of the arms and face. At the age of 16, he received a diagnosis of epilepsy with generalized tonic–clonic seizures, and he was treated with valproic acid. Despite treatment adherence, seizures persisted, so levetiracetam was administered as an add-on therapy. From the age of 17, the patient started experiencing hallucinations as well as emotional and behavioral disturbances, followed by the onset of cognitive decline, dysarthria, dysphagia, and gait impairment. Upon admission, the patient was wheelchair bound, unable to communicate effectively, and needed complete assistance for the activities of daily living. Neurological examination revealed frequent myoclonic twitches, severe language impairment, excessive daytime sleepiness, and limb ataxia with impaired coordination and balance. Social interaction with the hospital personnel was difficult, as the patient refused to eat or take medication in the absence of parents. Examination of other systems showed no pathological findings. Partial seizure control was achieved when a third drug, perampanel, was added to the regimen. During hospitalization, the patient had episodes of visual psychotic hallucinations, that were managed and put under control by administering olanzapine.

Routine blood tests were unremarkable. Both serum and cerebrospinal fluid examinations ruled out the hypotheses of infectious or immune-driven origins. We conducted Magnetic Resonance Imaging (MRI) of the brain and spine, along with electroencephalogram (EEG) tests (Figure 1, Appendix A). EEG showed bilateral, mainly synchronous, multifocal 3–4 Hertz (Hz) discharges, interrupted by brief phases (0.5–1 s) of increased spike frequency (up to 5–6 Hz) (Figure 1A,B). Brain MRI was normal (Figure 1C,D). After diazepam administration, a significant suppression of epileptiform activity was observed, with alternance of 3–4 Hz spike-waves and irregular theta activity.

In consideration of such a complex clinical picture including a severe, progressive, and drug-resistant myoclonic epilepsy, the patient underwent NGS analyses. Simultaneously, a muscle biopsy was performed.

## 3. Results

The muscle biopsy showed normally shaped fibers with a physiological variability in size, without splitting or internal nuclei (Figure 2A). Several myofibers displayed multiple small vacuoles, some with a fuchsinophilic border (Figure 2B). The specific Oil Red-O staining demonstrated a significant increase of lipid droplets (Figure 2C), which was also confirmed through electron microscopy (Figure 2D). The periodic acid–Schiff (PAS) reaction and ultrastructural examination also showed a very mild increase of intermyofibrillar glycogen. However, PAS staining with diastase (PAS-D) demonstrated the absence of polyglucosan accumulations typical of Lafora bodies (Figure 2E,F).

Clinical exome sequencing identified the presence of the homozygous c.137G>A, p.(Cys46Tyr) variant in the *EPM2B* gene (NM_198586.3), supporting the diagnosis of LD type 2B. The presence of this variant was also confirmed by Sanger Sequencing of the proband as well as his healthy heterozygous parents, suggesting a biparental origin (Appendix A). The variant is currently classified as likely pathogenic, according to criteria set forth by the American College of Medical Genetics and Genomics (ACMG) [18,19]. Genetic counseling was provided accordingly.

## 4. Discussion

LD is a catastrophic form of Progressive Myoclonic Epilepsy, with a common onset during late childhood or teenage years [20], often characterized by symptoms such as migraines, deteriorating academic outcomes, and pronounced seizures. Early on, individuals with LD tend to exhibit normal behaviors, though a minority of them might show signs of learning difficulties. Progression is marked by tonic–clonic seizures and is subsequently accompanied by abrupt, widespread muscle spasms in the extremities and facial region. Transient visual disturbances and illusionary experiences are also noted. A sharp decline in cognitive abilities and memory typically becomes evident between 2–6 years following the onset of the disease [6]. Additionally, affected individuals may be dealing with issues like disorientation, speech and language challenges, compromised decision making, and mood disorders. The involvement of the cerebellum might lead to difficulties with movement synchronization and balance [21]. Tragically, the disease often culminates in death, approximately a decade after the emergence of symptoms, primarily due to severe complications like intense seizures or respiratory infections [16,22].

In the early stages, brain MR scans might not show any abnormalities, but as LD advances, some patients may present with widespread brain degeneration. However, some others may continue to present standard MRI findings. EEG abnormalities often arise before clinical signs become tangible. EEG patterns may first appear nearly typical or showcase reduced background rhythms, with isolated or generalized sudden epileptic episodes. As the condition evolves, a pronounced decline in background activity is observed, complemented by recurrent epileptic episodes [16].

Pathologically, polyglucosan deposits can be found in all glycogen-producing cells, such as the brain, liver, muscles, and sweat glands [23]. Before the discovery of LD genes in the late 20th century, the diagnosis of LD was mainly based on the identification of LBs in brain, liver, skeletal muscle, and skin biopsies. In particular, the axillary skin biopsy frequently shows PAS-positive inclusions within the cells of the sweat ducts. However, the chance of false-positive [24] and false-negative [25] findings might limit the reliability of the axillary skin biopsy as an exclusive diagnostic technique. As a result, the identification of biallelic mutations in either the *EPM2A* or *EPM2B* gene is now required in order to confirm the diagnosis of LD [26].

The physiological functions of laforin and malin proteins and the consequences of mutations in their respective genes have been extensively studied [2]; however, the root causes behind the build-up of insoluble glycogen in LD are not entirely clear yet. In earlier studies using LD mouse models, increased phosphate levels were observed in the glycogen of LD muscle and brain tissues [27,28,29]. While this is expected in laforin-lacking Epm2a−/− mice [30], it is noteworthy that malin-lacking mice (Epm2b−/−) present with glycogen phosphate levels sitting between those of wild-type and *EPM2A*−/− mice [31,32]. Such an unusual rise in phosphate may contribute to the development of insoluble glycogen and to its subsequent deposition in LD tissues [32].

More recent studies led by Sullivan et al. [33] provided new insights into the foundational mechanisms of insoluble glycogen formation, and into the impacts of deficiencies in laforin, malin, and GBE1 (glycogen branching enzyme) on glycogen solubility. Through LD mouse models of both Epm2a−/− and Epm2b−/−, the research identified the presence of PAS-positive, diastase-resistant polyglucosan structures and an accumulation of glycogen in both cerebral and muscular tissues. This coincided with increased phosphorylation and branching anomalies. Moreover, their work showed that excessive glycogen phosphorylation is not the sole cause of polyglucosan formation when laforin-lacking phosphatase activity is present. The formation of these structures was, in fact, halted in LD mouse models upon the genetic removal of protein targeting glycogen (PTG) or glycogen synthase (GYS, chain-elongating).

In the pre-genetic era, LBs were searched for in the skeletal muscles of LD patients, in order to support the clinical diagnosis [34,35,36]. However, the lack of a molecular diagnosis in these patients prevents the drawing of any conclusions about the correlation between the specific molecular defect and the observed pathological alterations [36,37,38]. In addition, after the discovery of the causative genes for Lafora disease, the use of biopsy as a diagnostic confirmation method, particularly when applied to skeletal muscle, has diminished. Therefore, recent literature lacks the detailed description of histological muscle findings associated with specific gene mutations. When histological features are described, (i) they refer to skin biopsy [25] or (ii) authors do not specify the analyzed tissue [39]. A single study described the muscle biopsy of a LD patient harboring a homozygous *EPM2A* mutation [40]: histological analysis highlighted uneven muscle fiber dimensions, with most of the fibers showing considerable sarcoplasmic vacuolization; whilePAS staining identified the presence of PAS-positive polyglucosan aggregates.

In our specific case study, the muscle biopsy showed the presence of multiple tiny vacuoles within myofibers. However, specific stainings revealed that these structures were consistent with lipid, rather than polyglucosan, accumulations. Atypical intraperoxisomal polysaccharide accumulations had been previously observed in the skeletal muscle of a LD patient [34] and interpreted as the result of altered autophagy mechanisms (lipophagy), due to excessive glycogen accumulation [41]. In addition, Neville and colleagues in 1974 [36] revealed that muscle biopsies of two patients suffering from LD exhibited a notable stippling pattern in muscle fibers which, under ultrastructural analysis, was identified as small, membrane-bound clusters containing densely osmiophilic granules. Some reports also described the presence of lipofuscin granules in neurons of Epm2a or Epm2b knockout mice [30,42,43].

The specific relationship between such lipid storages and glycogen dysfunction remains unclear. Hence, we cannot exclude the alternative hypothesis of a secondary deficiency of muscle carnitine, which may possibly concur with the formation of these lipid deposits. This can also be seen in some patients undergoing chronic treatment with valproic acid [44,45].

Our histological findings contribute, indeed, to the recognition that LD muscle tissues can exhibit accumulations beyond LBs. The laforin–malin complex, beside the regulation of glycogen, is involved in multiple cellular pathways, including endoplasmic reticulum stress responses and protein quality controls [46,47]. Therefore, it is plausible that additional autophagic substrates, such as proteins and lipids, can accumulate in Lafora tissues, contributing to pathological changes. This observation could be valuable for other clinicians, aiding in the diagnostic pathway of myoclonic epilepsy, when considering mitochondrial myoclonic epilepsy as a potential differential diagnosis. On the other hand, our case suggests that the absence of LBs in muscle tissue, as observed by others in skin fibroblasts [25], does not exclude the diagnosis of LD, thus strengthening the advantage of genetic analysis to achieve an unambiguous confirmation of this disorder. In our study, the description of this peculiar histological aspect is related to a single patient and a single tissue (skeletal muscle; while skin biopsy was not available), limiting the general validity of these findings.

Theoretically, the synthesis and degradation pathways of glycogen are the same in all body cells, except for neurons, which rely less on these pathways compared to organs like liver, kidneys, and muscles. Neurons are somewhat more dependent on astrocytes, both in this aspect and for other processes [48]. As a result, if LBs were merely a consequence of faulty glycogen degradation, they should be readily found in tissues and cells across all glycogen-related disorders, such as those related to mutations in Glycogenin-1, Glycogen synthase, and Glycogen branching and debranching enzymes [49]. However, the data do not align with this notion. For instance, they are absent in cases of debranching disorders. In peripheral tissues of patients affected with Adult Polyglucosan Body Myopathy, these bodies are more prominent in peripheral nerves (axons) than in muscles, where they are inconstantly observed [50]. On the other hand, in neonatal branching disorders, the polyglucosan bodies are widespread and exhibit distinct characteristics from the typical ones [51]. Clearly, there might be other factors influencing the varying prevalence of polyglucosan bodies in different cell types.

Although confirming the presence of LBs during tissue biopsies can aid in the diagnostic process of LD, distinguishing them from normal cell granules can be challenging [24]. For this reason, genetic testing stands as the key approach to definitively diagnose LD, ensuring accurate identification of the underlying genetic mutations associated with the disease [2].

We detected the *EPM2B* variant c.137G>A, p.(Cys46Tyr), as the likely cause of the LD diagnosed in our patient. This variant was previously reported in the literature as associated with LD, in a compound heterozygous status with the c.205C>G, p.(Pro69Ala) variant [9]. Such a mutation is known, through in vitro functional assaying, to impair the interaction between malin and laforin, thus preventing the proteasomal degradation of the glycogen synthesis regulator protein PTG and resulting in glycogen accumulation [9]. Although pathological, the genotype observed in our patient is expected to allow the production of a residual amount of malin, which may be the factor underlying the limited impact on our patient’s muscle. Conversely, the knockout Epm2b−/− mouse model—displaying the complete genetic abrogation of *EPM2B* and its protein product—shows the diffuse presence of LBs in multiple tissues, including muscle [33,52].

Distinguishing LD patients with *EPM2A* mutations from those harboring *EPM2B* variants is a clinical challenge, making genetic testing imperative in order to determine the specific LD variant. As of now, around 150 unique mutations in the *EPM2A* and *EPM2B* genes have been identified across over 250 LD patients and/or families [16,17]. Half of these are missense mutations, with deletions accounting for roughly a quarter [15]. The broad array of allele variations and the common presence of compound heterozygotes in multiple combinations complicate the task of drawing clear connections between genotypes and phenotypes [14,53].

*EPM2A* and *EPM2B* are ubiquitously expressed, but regional differences in their expression exist and are observed even in glycogen tissues. Indeed, laforin is more abundant in the muscle compared to the brain, whereas the levels of malin are low in myofibers and higher in neuronal cells. Despite the recognized role of the laforin–malin complex in glycogen synthesis regulation, it is likely that these proteins also display tissue specific roles, resulting in differential impacts of their absence in different cell types. A deeper dive into research is needed to establish the relationship between genetic anomalies and clinical manifestations.

Initial theories posited that LD arising from the *EPM2B* gene mutation had a delayed onset and less severe symptoms compared to LD linked to the *EPM2A* gene mutation [54]. Yet, more contemporary research focusing on *EPM2B*-associated LD suggests that the condition can also progress rapidly. Conversely, there have been instances where *EPM2B* mutations were correlated with both gradual and swift disease advancements [15,25,55]. Recently, Pondrelli and co-authors [17] performed a systematic review of molecularly confirmed LD patients and demonstrated the existence of a correlation between the genotype (variant type) and the clinical course of the disease (prognosis). They stated that patients carrying biallelic protein-truncating mutations in the *NHLRC1* gene (PT/PT genotype) showed worse advancement and prognosis in terms of age of onset, survival, and autonomy impairment when compared to patients presenting with biallelic missense variants (MS/MS genotype).

Regarding our specific case study, the patient harbored a biallelic missense variant in the *NHLRC1* gene, showed the first symptoms before the age of 18 years, and underwent a gradual yet consistent decline, culminating in significant neurological impairment around at 20 years of age. In fact, by the diagnostic stage, he was nearly unable to walk or communicate verbally. Collectively, our observations support the notion that *EPM2B* mutations can correlate with a range of progression rates, from gradual to rapid, and that biallelic missense variants can be also associated with a severe disease course.

As far as LD treatments are concerned, no effective LD therapy has been approved yet. Although antiepileptic drugs (AEDs) can partially alleviate myoclonus and seizures, they have limited impact on the progression of cognitive and behavioral symptoms, as well as on survival [2,56,57]. Studies have shown a more favorable response to perampanel compared to other AEDs [57,58,59,60]. In light of the previous studies, our patient was also put on perampanel, resulting in some improvement of the epileptic symptoms.

Beside symptomatic management, targeted and curative LD therapies are still lacking, even if progress at the preclinical stage has been made [13] (Figure 3).

Gene therapy holds significant promise in addressing debilitating genetic disorders, and it has shown notable advancements in the treatment of hereditary diseases. Considering that only two genes (*EPM2A* or *EPM2B*) play a role in LD, it positions the disease as a suitable target for gene replacement strategies. Adeno-Associated Viruses (AAVs) have garnered attention as the optimal and most reliable gene delivery method for LD, as well as other disorders affecting the central nervous system (CNS). This is largely due to their benign characteristics and remarkable efficiency in gene transfer [61,62,63]. Nevertheless, there are some challenges for successful AAV-based treatments to overcome. One difficulty is represented by the presence of the blood–brain barrier (BBB), which poses some obstacles to prevent AAV particles from accessing the brain. Direct injection of AAVs into the cerebrospinal fluid is being explored, in order to bypass the BBB and minimize systemic side effects [62,64]. Another area of research focuses on engineering viral capsids with enhanced BBB penetration capabilities. While progress has been made, achieving sufficient transduction efficiency throughout the brain remains a hurdle [64,65]. Early intervention and the identification of suitable strategies are crucial to halt neurodegeneration and prevent irreversible damage to the CNS in LD cases. Human safety trials for gene therapy in LD are currently in the planning stage, offering hope for those affected by this devastating disease.

In an innovative study conducted by Vemana et al. [66], a promising approach using EPM2A-loaded DLinDMA (a type of ionizable cationic lipid) lipoplexes was investigated. The careful formulation and characterization of DLinDMA and DOTAP liposomes resulted in cationic liposomes with favorable physicochemical properties. These nanosized DLinDMA liposomes have demonstrated exceptional transfection efficiency, while exhibiting minimal hemolysis and cytotoxicity. The successful expression of the desired protein, laforin, was confirmed through western blotting, and the biological activity of laforin was demonstrated in glucan phosphatase assays. This groundbreaking preclinical study signifies a crucial advancement in the utilization of cationic lipoplexes containing plasmid DNA, holding great potential for the treatment of rare genetic disorders such as LD. The restoration of transgenic malin, which is associated with LB degradation, has also shown promising results, leading to a halt in LB accumulation and an improvement in neuroinflammation [67].

Alternative therapeutic strategies for LD include the downregulation of glycogen synthesis by targeting glycogen synthase (GS), as well as the degradation of LBs [13]. Inhibiting GS activity has shown promising results in LD mouse models, preventing LB formation, and improving neurological symptoms [68,69]. Studies have demonstrated that a 50% decrease in GS activity can slow down disease progression [68,69]. Transgenic animal studies have revealed that reducing GS activity—by introducing a mutation which produces half of the normal enzyme—is sufficient to provide the necessary glycogen for survival without causing LD [70].

Targeting GS can be achieved at the DNA, RNA, or protein levels. The CRISPR-Cas9 system offers a promising method for DNA manipulation [71], with the potential to knock out the brain-expressed isoform of GS (encoded by GYS1). Delivery of CRISPR-Cas9 targeting GYS1 with the use of AAV vectors has demonstrated efficacy in reducing LB accumulation and neuroinflammation in LD mouse models [72]. The oligonucleotide-based therapeutics and RNA interference (RNAi) methods provide effective approaches for targeting GS. Antisense oligonucleotides (ASOs), such as Gys1-ASO, have been utilized to prevent LB formation in young mice and halt further LB accumulation in older LD mouse models [73]. These approaches—whether targeting DNA, RNA, or proteins—offer viable options for LD treatment. Ongoing research aims to optimize delivery methods and maximize the therapeutic potential of these interventions. However, experimental and genetic data suggest caution in the translatability of this substrate-reduction approach. Indeed, the genetic inactivation of GYS1 was found to be beneficial in animal models of glycogen storage disorder (such as the GAA-KO mouse, the animal model of Pompe disease [74]), but detrimental for wild-type mice [75]. Inactivating mutations of GYS1 are responsible for glycogen storage disease type 0, which is associated with muscle glycogen synthase deficiency and leads to pediatric cardiac arrest [76]. In addition, while the excessive accumulation of glycogen is toxic for neurons, the reduction of its levels is also potentially hazardous. Brain glycogen is protective under stress and pathological conditions, and it is likely involved in the preservation of learning capacity [48]. Therefore, a therapy aiming to modulate the homeostasis of glycogen should focus on maintaining its levels at the correct physiological interval.

Targeting the degradation of LBs is also a therapeutic approach for treating LD. VAL-0417, an antibody–enzyme fusion, has shown promising results both in vitro and in vivo, by degrading LBs and reducing LB accumulation in a mouse model of LD. This precision therapy has the potential to alleviate the physiological effects associated with LB accumulation [77,78,79].

In addition to these specific strategies, dietary adjustments, like the adoption of a ketogenic diet, might be considered as complementary treatments. A recent investigation by Israelian et al. [80] highlighted the potential of the ketogenic diet in minimizing unusual glycogen build-up in an LD mouse model. The authors advised beginning the diet concurrently with an LD diagnosis; they also emphasized the importance of a synchronized global clinical trial, to pinpoint the exact impact of this diet on individuals with LD.

Another approach involves repurposing existing drugs to target LD at a pathophysiological level. Metformin, commonly used to manage diabetes, has shown promise in reducing the accumulation of polyglucosans and polyubiquitin protein aggregates, as well as decreasing seizure susceptibility in LD mouse models [56,81]. In a groundbreaking study by Berthier et al. [82], metformin significantly reduced polyglucosan accumulation, polyubiquitin aggregates, neuronal loss, and reactive astrogliosis in the brain. It also exhibited remarkable anticonvulsant properties, reducing seizure frequency and duration, and preventing mortality induced by pro-convulsive agents [81]. Metformin was subsequently classified as an orphan drug for LD treatment by the EMA in 2016 and the FDA in 2017. Nevertheless, clinical outcomes remain uncertain, as a study with 10 LD patients yielded inconclusive results [56]. These were likely due to the advanced disease stages of the enrolled patients, underlining once more the importance of an early intervention [56]. More positive was, instead, the outcome of a recent study by Burgos et al. [83], which showed the potential benefits of early treatment with metformin in LD. A total of 18 LD patients were enrolled, with 8 receiving metformin and 10 remaining untreated. The patients on metformin exhibited a significantly slower progression of the disease compared to the untreated individuals.

Repurposing drugs with anti-inflammatory properties also holds promise for LD treatment. For example, Molla et al. [84] investigated the potential therapeutic effects of propranolol and epigallocatechin gallate (EGCG) in an LD mouse model, highlighting the effectiveness of inflammation modulators as novel treatment options.

## 5. Conclusions

In conclusion, both advanced therapies and repurposed drugs offer potential avenues for LD treatment. Early intervention and treatment initiation may play a crucial role in improving outcomes for LD patients. Further research and clinical studies are needed to fully explore the effectiveness, safety, and long-term benefits of these treatment approaches to LD.

Overall, this report emphasizes the importance of considering LD during the differential diagnosis of adolescents with progressive myoclonic epilepsy, cognitive decline, and diffuse neurological deterioration. Lafora disease carries a severe prognosis, rendering patients fully dependent and bed-bound, with death typically occurring within a decade of the onset of symptoms. However, recent advances in the diagnosis and treatment of LD, including a better understanding of its clinical features and groundbreaking gene therapy research, provide hope for better future management of this life-threatening disorder.

## Figures and Tables

**Figure 1 brainsci-13-01679-f001:**
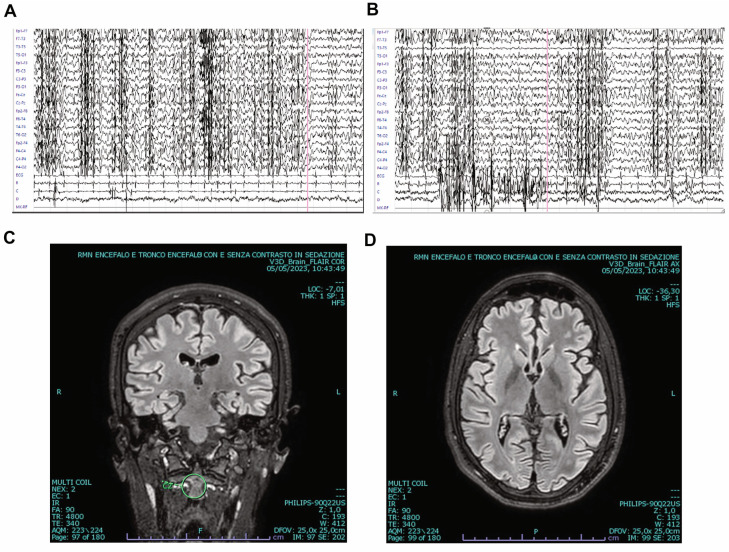
Electroencephalogram (EEG) and brain Magnetic Resonance Imaging (MRI) of our patient affected with Lafora disease. (**A**,**B**) EEG performed at admission showing epileptic discharges in both hemispheres (see also Appendix A). (**C**,**D**) Coronal (**C**) and transverse (**D**) brain MRI scans revealing normal morphology.

**Figure 2 brainsci-13-01679-f002:**
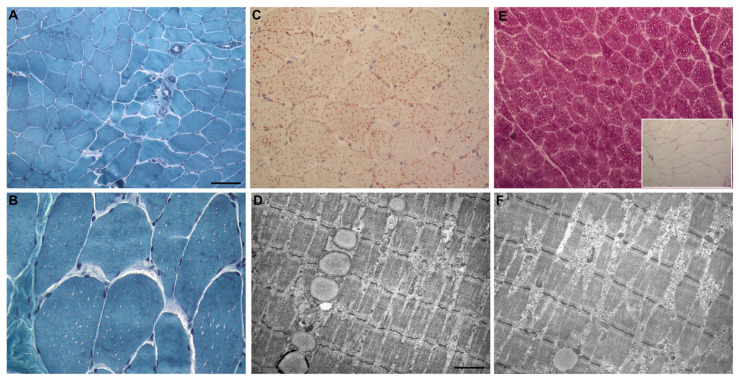
Myopathological features of our patient affected with Lafora disease. (**A**): Gomori’s trichrome stain: muscle biopsy showing normally shaped fibers with a physiological variability in size. (**B**): Gomori’s trichrome stain: multiple small vacuoles in numerous myofibers, compatible with lipid droplets. (**C**): Oil Red-O: increased lipid content. (**D**): Ultrastructural picture showing a lipid droplet chain. (**E**): Periodic acid–Schiff (PAS)-positive reaction, and PAS-Diastase (PAS-D) reaction (inset). (**F**): Ultrastructural picture showing small collection of intermyofibrillar glycogen. Scale bar. (**A**,**E**): 50 μm; (**B**,**C**): 25 μm; (**D**,**F**): 1.43 μm.

**Figure 3 brainsci-13-01679-f003:**
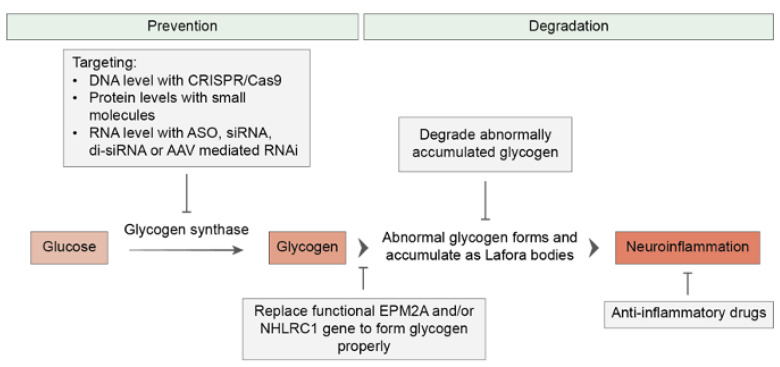
Overview of therapeutic strategies for Lafora disease. Therapeutic approaches to LD can be classified into two main groups. The first group aims to prevent or arrest the disease’s progression by targeting glycogen synthase or replacing the *EPM2A* and *EPM2B* (*NHLRC1*) genes. The second group of therapeutics aims to alleviate the existing disease by degrading Lafora bodies and reducing LD-related neuroinflammation.

## Data Availability

The data presented in this study are not publicly available due to their containing information that could compromise the privacy of the patient.

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
