# Peer review of "Lafora Disease: A Case Report and Evolving Treatment Advancements"

_brainsci, 2023, doi:10.3390/brainsci13121679_

Round 1

Reviewer 1 Report

Comments and Suggestions for Authors

This is a case report and review article on Lafora disease. Case report of a 20-year-old male patient with progressive myoclonic epilepsy and cognitive decline. 

1. The case report and literature review sections are scientifically sound. The treatment update is also relevant and timely. 

2. The overall innovation is limited - mostly a summation of existing research without new insights.

3. The logic flow is reasonable. Language is smooth.

4. I suggest shortening the text, reducing unnecessary references, improving figures, and adding a "Limitations" section.

Overall this is a comprehensive review of Lafora disease but lacks novelty. 

Comments on the Quality of English Language

Overall, the quality of the English language in the manuscript is good. Here are my comments on the language use:

- The language flows smoothly and is easy to understand. Technical terms are used appropriately. 

- Sentence structure and phrasing are generally clear and concise. There are no obvious grammatical errors.

- The tone and style are appropriate for a scientific publication. The language is formal, objective, and avoids idiomatic expressions.

- There is some repetition of ideas/phrases in a few places. The text could be tightened further to avoid redundancy.

- Some sentences are long and could be broken down. More varied sentence structure could enhance readability. 

- There are a few typos and minor errors that need fixing (eg. missing words, incorrect capitalization). But these are infrequent.

Reviewer 2 Report

Comments and Suggestions for Authors

The manuscript by Aggradi and colleagues is well-written and will be of interest to the glycogen storage disease field. I found it very enjoyable to read, scientifically accurate, and informative. I would suggest some edits prior to publication

1) The authors present a VERY unique case of Lafora Disease. The patient has a disease causing mutation in EPM2B yet the biopsy is Lafora body negative. It is HIGHLY unusual to find these two disconnected. Instead of Lafora bodies, the authors report what appears to be lipid aggregates. Given this unique finding, the paper should be about the unique case. The authors should reframe the manuscript stating that the patient has EPM2B mutations and a clinical presentation of LD yet there are no LBs and instead there are lipid accumulations. Then in the discussion the authors should propose a connection between lipids and glycogen and suggest that fundamental scientists investigate that link.

2) I would suggest a change in the title to more accurately reflect what the paper is about. Something like, "LD: case report and evolving treatment advancements"

3)Given the authors findings, they should reference and include some verbiage that others have reported that LBs are surrounded by lipofuscin in the muscle biopsy of a LD patient (PMID: 4133238, the one that I had recalled). Also, both lafoin KO by Ganesh (PMID:12019206), and malin KO created by Santiago Cordoba’s group (PMID: 22186026) also show lipofuscin in the brain.  

4) The authors claim that there are no LBs. However, they should at least see glycogen in a muscle biopsy. In order to be convinced that the PAS staining is working, they should at least show PAS staining in muscle to see glycogen.

Is this really the only one? This sentence is confusing. I think what they are trying to refer here is the only report in a homozygous EPM2A mutation patient showing vacuolization in the muscle biopsy sample, as I find papers with PAS staining in the muscle biopsy and homozygous mutation in EPM2A in LD.  Please clarify.

A few references- 

Muscle biopsy- 

  1. PMID: 4133238 
  2. PMID: 1654400 
  3. PMID: 22961547 

Homozygous EPM2A mutations 

  1. PMID: 32587944 
  2. PMID: 25246353 
  3. PMID: 20738377

"In the literature, there is one study documenting the muscle biopsy from an LD patient 184 harboring a homozygous EPM2A mutation [36]."

This sentence should be softened a bit given that the authors do no provide controls. If they wish to state this then they should stain the tissue with a lipid agent to determine that it is indeed a lipid. Otherwise they should change the word "recognition" to "suggests."

"Our histological findings contribute indeed to the recognition that LD muscle tissues can exhibit accumulations beyond LBs."

There is a new study from a clinical group that predicts pathogenic variants in LD patients. They should reference this work. 

Prognostic value of pathogenic variants in Lafora Disease: systematic review and meta-analysis of patient-level data. Orphaned Journal of Rare Diseases

Round 2

Reviewer 1 Report

Comments and Suggestions for Authors

The modification has been accepted.